# Precardiac deletion of Numb and Numblike reveals renewal of cardiac progenitors

Lincoln T Shenje[1,2,3†], Peter Andersen[1,3†], Hideki Uosaki[1,3], Laviel Fernandez[1], Peter P Rainer[1,4], Gun-sik Cho[1,3], Dong-ik Lee[1], Weimin Zhong[5], Richard P Harvey[6], David A Kass[1], Chulan Kwon[1,3*]

[1]Division of Cardiology, Department of Medicine, Johns Hopkins University, Baltimore, United States; [2]The Knight Cardiovascular Institute, Oregon Health & Science Universtiy, Portland, United States; [3]Institute for Cell Engineering, Johns Hopkins University, Baltimore, United States; [4]Division of Cardiology, Medical University of Graz, Graz, Austria; [5]Department of Molecular, Cellular and Developmental Biology, Yale University, New Haven, United States; [6]Developmental and Stem Cell Biology Division, Victor Chang Cardiac Research Institute, Darlinghurst, Australia

**Abstract** Cardiac progenitor cells (CPCs) must control their number and fate to sustain the rapid heart growth during development, yet the intrinsic factors and environment governing these processes remain unclear. Here, we show that deletion of the ancient cell-fate regulator Numb (Nb) and its homologue Numblike (Nbl) depletes CPCs in second pharyngeal arches (PA2s) and is associated with an atrophic heart. With histological, flow cytometric and functional analyses, we find that CPCs remain undifferentiated and expansive in the PA2, but differentiate into cardiac cells as they exit the arch. Tracing of Nb- and Nbl-deficient CPCs by lineage-specific mosaicism reveals that the CPCs normally populate in the PA2, but lose their expansion potential in the PA2. These findings demonstrate that Nb and Nbl are intrinsic factors crucial for the renewal of CPCs in the PA2 and that the PA2 serves as a microenvironment for their expansion.

*For correspondence:
ckwon13@jhmi.edu

†These authors contributed
equally to this work

Competing interests: The authors declare that no competing interests exist.

## Introduction

Embryonic cardiac progenitor cells (CPCs), identified from early embryos or differentiating pluripotent stem cells, hold tremendous regenerative potential with their unique capability to expand and differentiate into nearly all cell types of the heart (*Parmacek and Epstein, 2005*; *Kattman et al., 2006*; *Moretti et al., 2006*; *Kwon et al., 2007*). Over the past decade, significant progress in developmental cardiology led to the identification of CPC markers and lineages (*Cai et al., 2003*; *Kattman et al., 2006*; *Moretti et al., 2006*; *Kwon et al., 2009*). However, CPCs are highly heterogeneous and it is unknown if they can undergo self-renewal without differentiation. Consequently, understanding the precise mechanisms of CPC self-renewal and maintenance remains a fundamental challenge.

Cardiogenesis initiates as the basic helix-loop-helix protein mesoderm posterior 1 (Mesp1) is transiently expressed in the nascent mesoderm during gastrulation (*Saga et al., 1996*). Mesp1+ cells migrate anteriorly and form the first heart field (FHF) and second heart field (SHF) (*Saga et al., 2000*). The FHF gives rise to the atria and left ventricle (LV), whereas the outflow tract (OT), right ventricle (RV) and some of atria are derived from the SHF (*Buckingham et al., 2005*). Before myocardialization, subsets of Mesp1 progeny express CPC markers including Islet1 (Isl1), fetal liver kinase 1 (Flk1), Nkx2.5, or myocyte-specific enhancer factor 2c (Mef2c) in precardiac mesoderm (*Stanley et al., 2002*; *Cai*

**eLife digest** Human embryos contain cells called 'cardiac progenitor cells' that serve as the building blocks to make the heart. Cardiac progenitor cells, or CPCs for short, initially move into areas of the embryo called the first and second heart fields, and then undergo a change to become specific types of heart cells: such as cardiac muscle cells. However, it is not known if CPCs are maintained during the development of the heart.

Now, Shenje, Andersen et al. have shown that Numb and Numblike—two proteins that are needed for the development of nerve cells—are also involved in the development of the heart. Mouse embryos without the genes for Numb and Numblike failed to develop hearts normally; and these mutants also had fewer CPCs in the 'second pharyngeal arch': a part of the embryo that becomes the sides and front of the neck. Experiments on wild-type mice showed that the CPCs multiplied within this arch, and then changed into specific heart cells as they left this structure. Furthermore, mixing CPCs in a petri dish with cells taken from this arch encouraged the CPCs to multiply without changing into specific cell types.

To investigate the importance of these two proteins further, Shenje, Andersen et al. engineered 'chimeric' mice in which some CPCs contained the *Numb* and *Numblike* genes and other CPCs did not. In most of these chimeric mice, the hearts developed normally, but the CPCs without the *Numb* or *Numblike* genes failed to multiply in the second pharyngeal arch. This shows that these genes must be present within an individual CPC to regulate the multiplication of that cell within this arch.

By uncovering how problems with the maintenance of CPCs can lead to heart defects—a very common birth defect in humans—this work may lead to new ways to prevent or treat congenital heart disease. Furthermore, identifying the other factors or mechanisms that can allow the long-term maintenance of CPCs in the laboratory will be crucial for research into heart regeneration, and for CPC-based treatments to repair the heart.

*et al., 2003*; *Verzi et al., 2005*; *Kattman et al., 2006*). Isl1 and Flk1 expression is extinguished as CPCs adopt myocardial fates, but Nkx2.5 and Mef2c are continually expressed in cardiomyocytes (*Edmondson et al., 1994*; *Tanaka et al., 1999*). While CPCs expressing these markers have similar differentiation potential in vitro (*Kattman et al., 2006*; *Moretti et al., 2006*; *Wu et al., 2006*), it is unknown if a discrete population of stem cell-like CPCs exist to supply cells for cardiac growth and morphogenesis during development.

Numb and Numblike (Numbl)—mammalian Numb homologs sharing collinear topology and extensive sequence identity with functional redundancy—are evolutionarily conserved proteins that are required for the self-renewal of neural progenitors and mediate asymmetric cell divisions in various contexts of cell fate decisions (*Zhong et al., 1997*; *Petersen et al., 2002*, *2004*; *Roegiers and Jan, 2004*), but their role in CPC development has not been explored. In the current study, we sought to identify and investigate CPCs affected by Numb and Numbl. By taking combinatorial approaches, we demonstrate that Mesp1$^+$ progenitor-derived Isl1$^+$ Nkx2.5$^-$ cells renew and expand without cardiac differentiation in the second pharyngeal arch (PA2) and that PA2 serves as their microenvironment during mammalian heart development.

## Results

### Numb and Numbl are required for heart development

*Numb* is expressed ubiquitously in developing mouse embryos (*Zhong et al., 1997*; *Jory et al., 2009*; *Figure 1—figure supplement 1*). To quantitatively examine the expression of *Numb* and *Numbl* in developing CPCs, we used the embryonic stem (ES) cell differentiation system that recapitulates early cardiogenesis (*Kattman et al., 2011*; *Van Vliet et al., 2012*; *Figure 1A*). *Numb* levels were relatively low at day 4, when *Mesp1* was induced, but upregulated at day 6, when *Isl1* appeared (*Figure 1B*). *Numbl* levels were also increased at day 6 (*Figure 1B*), implying that Numb and Numbl may have a role in CPC development after *Mesp1* induction. To test this possibility, we deleted *Numb* in Mesp1$^+$ progenitors, the earliest mesodermal progenitors giving rise to the entire heart and vasculature

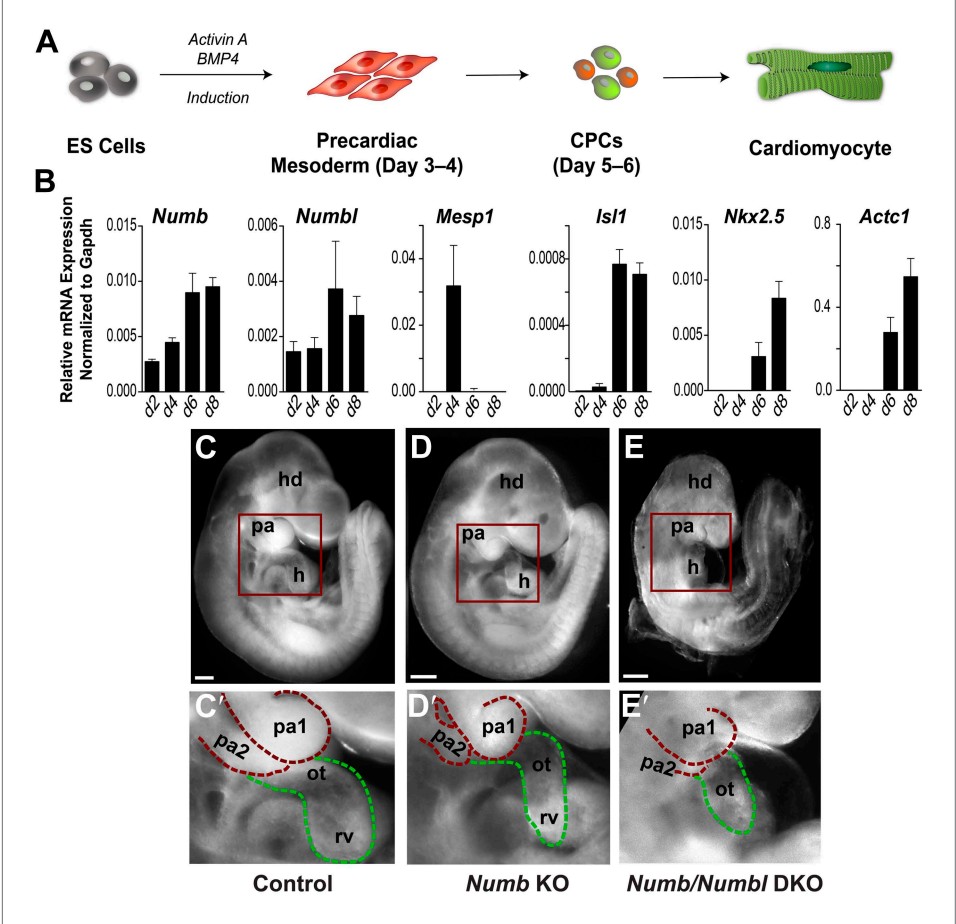

**Figure 1**. Numb and Numbl are required for PA2 and heart development. (**A**) Schema of cardiac differentiation in ES cell system. (**B**) Expression profiles of genes indicated during cardiac differentiation of ES cells. Gene expression was analyzed by qPCR. Data are mean ± SD; n = 4; d, day. (**C–E**) Lateral views of control (**C**), *Mesp1^Cre^; Numb^flox/flox^* (*Numb* KO, **D**), *Mesp1^Cre^; Numb^flox/flox^; Numbl^−/−^* (*Numb/Numbl* DKO, **E**) embryos. (**C′–E′**) Enlargement of boxed areas in (**C–E**), showing normal, hypoplastic or atrophic PA2 and heart in control (**C′**), *Numb* KO (**D′**) or *Numb/Numbl* DKO (**E′**) embryos, respectively. Pharyngeal arches (red) and outflow tract/right ventricle (green) are outlined in dashes. Scale bars, 150 μm. hd, head; pa, pharyngeal arch; h, heart; ot, outflow tract; ra, right atrium; rv, right ventricle.

The following figure supplements are available for figure 1:

**Figure supplement 1**. Numb is ubiquitously expressed in developing embryos.

---

(*Saga et al., 2000*), by crossing *Mesp1^Cre^* mice with *Numb^flox^* mice (*Zhong et al., 2000*). The deletion did not affect LV formation, but resulted in a hypoplastic OT/RV and PA2 (*Figure 1C,C′,D,D′*). While the phenotype appeared to be confined to the OT, RV, and PA2, the penetrance was variable. Since *Numbl*-null mice are viable and fertile, but Numbl can compensate for loss of Numb function (*Petersen et al., 2002*), we extinguished Numb and Numbl expression in Mesp1⁺ progenitors by deleting *Numb* in the *Numbl*-null background (*Petersen et al., 2002*). The resulting *Numb* and *Numbl* double knockout (*Numb/Numbl* DKO) embryos showed a hypoplastic OT/RV and PA2 with complete penetrance (n = 12 embryos, *Figure 1E,E′*) and uniformly lethal by embryonic day (E) 10.0. These data suggest that Numb and Numbl are required for the formation of the PA2 and OT/RV.

## Deletion of Numb and Numbl depletes Mesp1⁺ cell progeny in PA2

Based on the *Numb/Numbl* DKO phenotype, we focused on investigating Mesp1 progeny contributing to the PA2 and OT/RV. To visualize Mesp1 progeny affected by Numb and Numbl, we introduced a Cre reporter (Ai9) into the DKO mouse embryos, in which red fluorescent protein (RFP) permanently

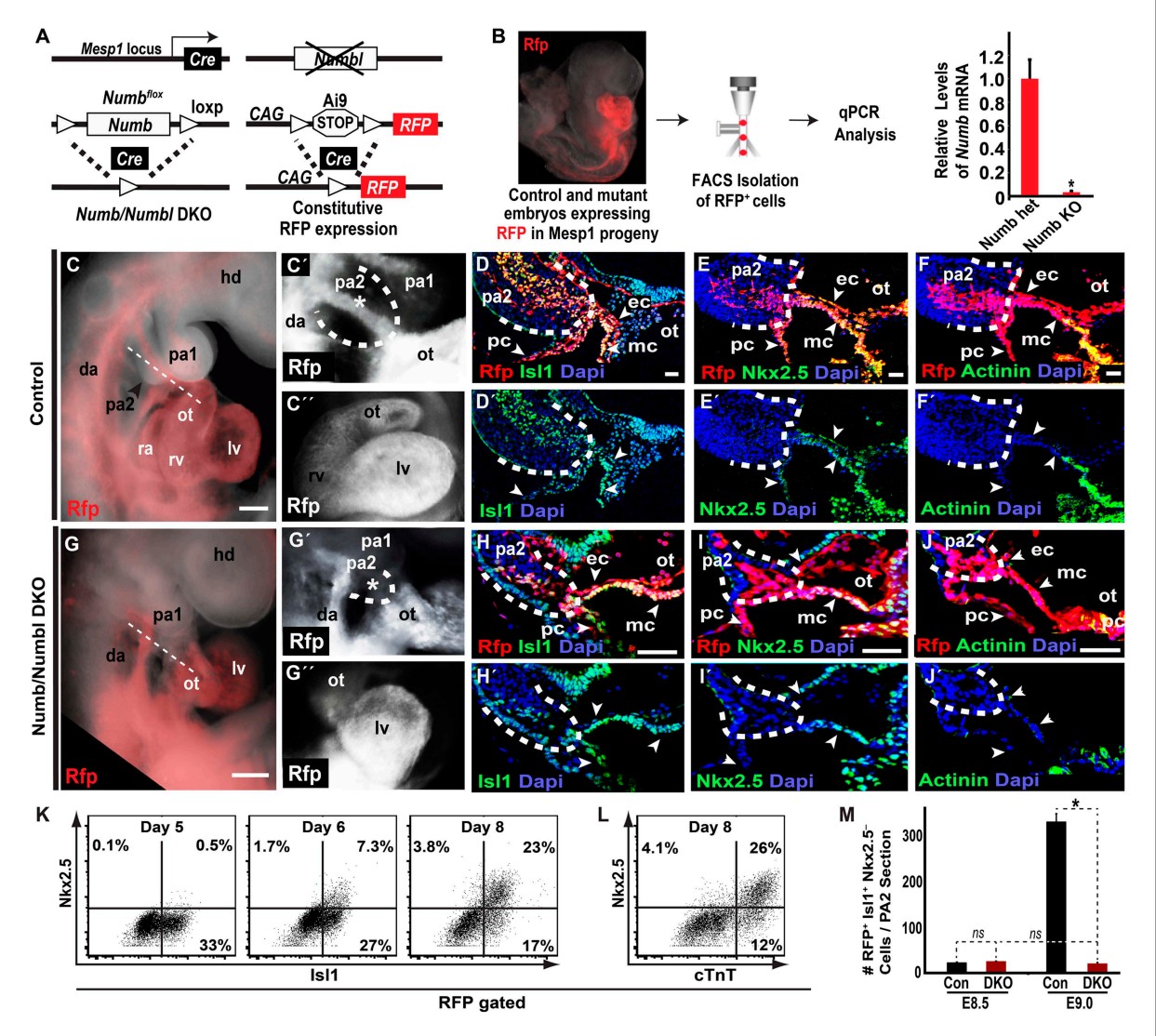

**Figure 2**. Deletion of Numb and Numbl depletes Mesp1+ progenitor-derived Isl1+ Nkx2.5– cells in PA2. (**A**) Generation of *Numb/Numbl* DKO embryos expressing RFP in Mesp1 progeny (*Mesp1^Cre^; Numb^flox/flox^; Numbl^–/–^; Ai9*). (**B**) Relative mRNA levels of *Numb* in RFP+ cells isolated from control and *Numb/Numbl* DKO embryos. Data are mean ± SD; n = 9. (**C**, **C'**, **G**, **G'**) Lateral views of Mesp1+ cell-traced control (**C** and **C'**) or *Numb/Numbl* DKO (**G** and **G'**) embryos analyzed at E9.0. RFP marks Mesp1 progeny. Control embryos show continuous RFP expression from second pharyngeal arch (PA2, outlined) to heart (asterisk, **C'**), but the arch (outlined) is severely underdeveloped in *Numb/Numbl* DKO embryos without noticeable RFP expression (asterisk, **G'**). (**C''** and **G''**) Frontal views of control (**C''**) or *Numb/Numbl* DKO (**G''**) heart. (**D–F'**), (**H–J'**), Representative confocal images of transverse sections through PA2 and outflow tract (OT) of control (**D–F'**) and *Numb/Numbl* DKO (**H–J'**) embryos. Cutting planes are shown in (**C**) and (**G**). Internal boundaries of PA2 are outlined in dashes. Dapi (blue) was used to counterstain the nuclei. (**K** and **L**) Representative plots of intracellular staining of RFP-gated cells. Isl1 and Nkx2.5 staining during day 5–8 (**K**) and cardiac troponin T (cTnT) and Nkx2.5 staining at day 8 (**L**). (**M**) Average number of RFP+ Isl1+ Mef2c–Nkx2.5– cells in PA2 section (12-micron) of indicated embryos and stages. Data are mean ± SD; n = 5; *p<0.05; *ns*, not significant. p values were determined using the paired Student *t* test. Scale bars, 25 μm (**D–F** and **H–J**), 150 μm (**C** and **G**). hd, head; pa, pharyngeal arch; ot, outflow tract; da, dorsal aorta; fe, foregut endoderm; ra, right atrium; rv, right ventricle; lv, left ventricle; ec, endocardial layer; mc, myocardial layer; pc, pericardial layer.

The following figure supplements are available for figure 2:

**Figure supplement 1**. Numb/Numbl DKO embryos are grossly normal at E8.5.

**Figure supplement 2**. Mef2c expression in PA2 and OT of control and Numb/Numbl DKO embryos.

**Figure supplement 3**. Histogram of RFP + cell induction from Mesp-Cre; Ai9 ES Cells.

marks Mesp1[+] progenitors, and traced the RFP[+] cells (*Figure 2A*). The resulting RFP[+] cells showed a near complete deletion of *Numb*, confirmed by fluorescence-activated cell sorting (FACS) and quantitative PCR (qPCR) (*Figure 2B*). Interestingly, control and DKO littermates showed no noticeable morphological defects at E8.5 (8–12 somite stage) and were histologically indistinguishable (*Figure 2—figure supplement 1*). RFP[+] cells were normally expressed the transient CPC marker Isl1 (*Cai et al., 2003*) from the primordial PA2 to the bulbus cordis (BC), and there was no significant difference in the number of RFP[+] Isl1[+] cells or the percentage of RFP[+] Isl1[+] cells positive for the mitosis marker phospho-histone H3 (PH3) (*Figure 2—figure supplement 1B,B′,G,G′,K*, *Figure 2M*). The RFP[+] Isl1[+] cells in PA2 primordia were continuous with the endo-, myo- and peri-cardial cell layers of the heart tube, and expressed the cardiac transcription factors Nkx2.5 and Mef2c from the distal BC and the sarcomeric protein α-Actinin from the proximal BC in both control and mutant embryos (*Figure 2—figure supplement 1B–E′,G–J′*). Thus, *Numb/Numbl* DKO does not appear to affect CPC and heart development at E8.5.

By E9.0 (18–22 somite stage) the RFP[+] cells in the PA2 were continuous with the OT in control embryos, but were nearly absent in *Numb/Numbl* DKO embryos (*Figure 2C–C″,G–G″*). Histological analysis revealed that the PA2 contained a dense cluster of RFP[+] Isl1[+] cells in control embryos, but they were severely depleted in *Numb/Numbl* DKO embryos (*Figure 2D,D′,H,H′,M*). This suggests that Numb and Numbl are required for the formation of the RFP[+] Isl1[+] cell cluster in the PA2. Intriguingly, the RFP[+] Isl1[+] cells in the PA2 did not express the reported CPC genes Nkx2.5 and Mef2c (*Dodou et al., 2004*; *Wu et al., 2006*), but were continuous with the Nkx2.5[+] Mef2c[+] cells in the distal OT and thereafter the α-Actinin[+] cells in the proximal OT and the RV (*Figure 2E–F′,I–J′*, *Figure 2—figure supplement 2*). This suggested that the transition of the Mesp1 lineage descendant to cardiomyocytes may occur through (1) Isl1[+] Nkx2.5[–]/Mef2c[–] α-Actinin[–] cells, (2) Isl1[+] Nkx2.5[+]/Mef2c[+] α-Actinin[–] cells, (3) Isl1[–] Nkx2.5[+]/Mef2c[+] α-Actinin[+] cells. To test this possibility, we rederived ES cells expressing RFP in Mesp1[+] cells from *Mesp1[Cre]; Ai9* embryos, induced Mesp1[+] precardiac mesoderm (*Kattman et al., 2011*), and analyzed CPC differentiation by FACS. RFP[+] cells appeared on day 4 of differentiation and expressed Isl1 on day 5 (*Figure 2—figure supplement 3*, *Figure 2K*). The RFP[+] Isl1[+] cells started to express Nkx2.5 from day 6 and differentiated into cardiac troponin T[+] (cTnT[+]) myocytes (*Figure 2K,L*). This indicates Mesp1[+] progenitors are specified to Isl1[+] Nkx2.5[–] CPCs and transition to Nkx2.5[+]/cTnT[+] cells in an ES cell system as they differentiate into cardiac cells. Together, these data suggest that Numb and Numbl are required to form a dense population of Mesp1[+] cell-derived Isl1[+] Nkx2.5[–] cells in the PA2, which may give rise to Nkx2.5[+] cardiac cells.

## Mesp1 progeny expand in PA2 and migrate out to become heart cells

While PAs contain SHF progenitors (*Kelly et al., 2001*; *Xu et al., 2004*), it is unknown if Mesp1[+] cell-derived Isl1[+] Nkx2.5[–] cells in the PA2 give rise to cardiac cells. Based on the *Numb/Numbl* DKO phenotype, cardiac gene expression patterns, and ES cell data, we hypothesized that the Isl1[+] Nkx2.5[–] cells expand without differentiation in the PA2 and migrate out of the arch to differentiate into cardiac cells. To test this, we examined proliferating cells in cardiac regions of E9.0 embryos by performing whole-mount staining with 5-ethynyl-2′-deoxyuridine (EdU), a nucleoside analog of thymidine. Remarkably, EdU[+] cells were concentrated in the PA2 and rarely detected in other cardiac regions including the growing heart (*Figure 3A,B*). Consistently, about four percent of Mesp1[+] cell-derived (RFP[+]) Isl1[+] Nkx2.5[–] cells were PH3[+] in the PA2, whereas PH3[+] cells were nearly absent in RFP[+] Isl1[+/–] Nkx2.5[+] OT and α-Actinin[+] RV cells (*Figure 3C*). Since DNA analogues are commonly used to show actively dividing stem cells and their lineages (*Eisenhoffer et al., 2008*; *Snippert and Clevers, 2011*), we labeled the proliferating cells by administering a single pulse of EdU, and monitored EdU[+] cells in RFP[+] cells (*Figure 3D*). EdU[+] cells were first identified in the outer layer of the RFP[+] cell cluster in PA2 after 2 hr, but not in the OT/RV (*Figure 3D*). By 4–8 hr, the RFP[+] EdU[+] cells were abundant in the PA2 and OT, and eventually found in the RV after 24–48 hr of chase (*Figure 3D*). RFP[+] cells in the PA2 remained EdU[+] (*Figure 3D,E*), implying their renewal in the PA2. The purdurance of the EdU pulse, determined by a sequential injection of EdU and 5-bromo-2′-deoxyuridine (BrdU, another thymidine analog), was shorter than 4 hr (*Figure 3—figure supplement 1*). These data suggest that Mesp1[+] cell-derived Isl1[+] Nkx2.5[–] cells in the PA2 migrate out of the arch and differentiate into cardiac cells.

To directly test the potential of RFP[+] cells in the PA2, we labeled proliferating cells with EdU for 2 hr at E9.0, dissected out PA2s from embryos (asterisks, *Figure 3D*), and cultured the arches in 3D Matrigel culture. RFP[+] cells were monitored in real-time with fluorescent time-lapse movies. After 2 days, the cluster of RFP[+] cells expanded and exhibited multidirectional migration

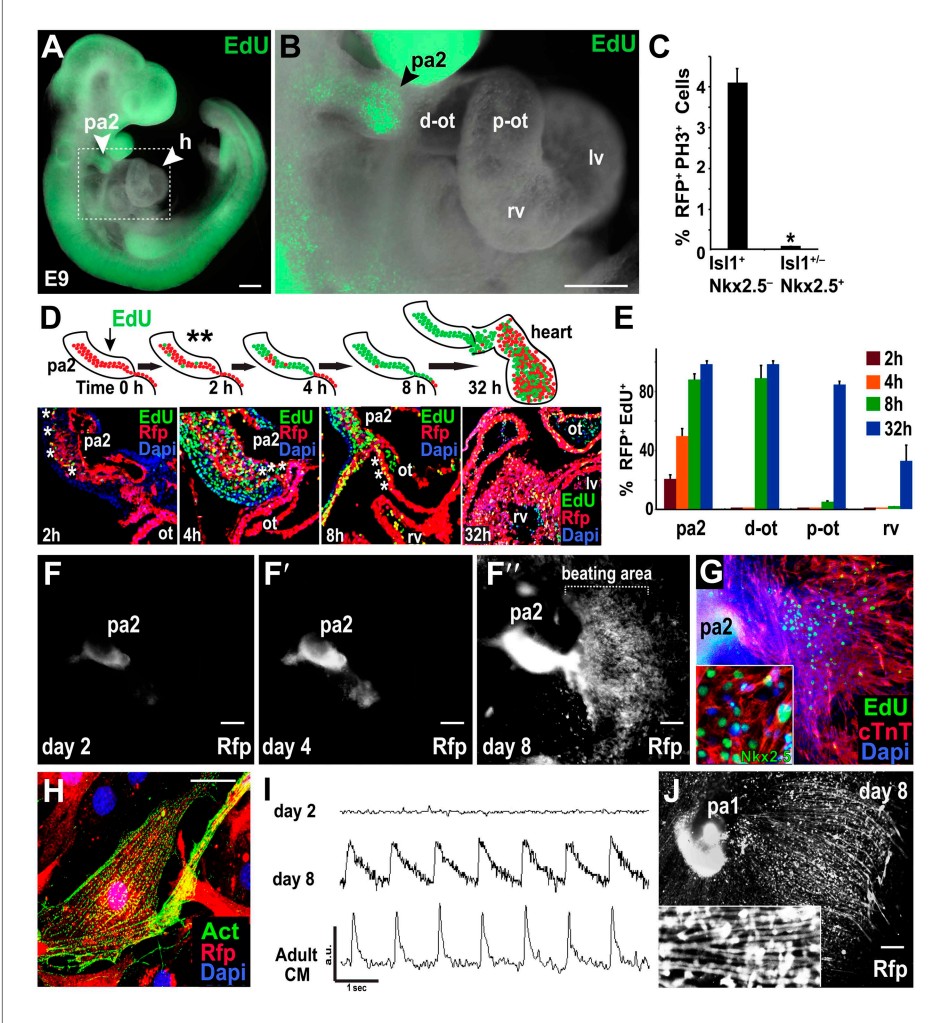

**Figure 3**. Mesp1+ progenitor-derived Isl1+ Nkx2.5– cells expand in PA2 and differentiate into Nkx2.5+ heart cells after leaving PA2. (**A**) Whole-mount view of EdU (green)-treated embryo at E9.0. (**B**) Enlargement of the boxed area in (**A**), showing enrichment or lack of EdU+ cells in the PA2 or heart region, respectively. (**C**) Percentage of RFP+ PH3+ cells in Isl1+ Mef2c–Nkx2.5– cells and Isl1+/– Mef2c+ Nkx2.5+ cells. Data are mean ± SD; n = 4. (**D**) EdU pulse experiment. Top, EdU experiment schema. PA2s were also dissected out for ex vivo culture after 2 hr (**\*\***). Bottom, progeny of EdU+ cells at 2, 4, 8, and 32 hr after single pulse of EdU injection at E9.0 demonstrating that Mesp1 progeny in PA2 proliferate and migrate to form the OT/RV. (**E**) Quantification of RFP+ EdU+ cell progeny shown in (**D**). Data are mean ± SD; n = 3. (**F–F"**), Cultured PA2 explants in 2D culture form a sheet of beating cardiomyocytes (see *Video 1*). (**G**) EdU-pulsed PA2-derived sheet of beating cells stained with cTnT, EdU, or Nkx2.5 (inset). (**H**) Confocal image of RFP+ cells migrated from PA2 stained with cardiac α-actinin (Act). (**I**) Intracellular Ca2+ transients from day 2 and day 8 PA2 explants and adult cardiomyocyte (CM). a.u., arbitrary unit. (**J**) Cultured PA1 explants form myotube-like cells. Inset shows a magnified view. \*p<0.05. D, day; Heart, E10.5 embryonic heart; *ACTC1*, actin, alpha, cardiac muscle 1. Dapi (blue) was used to counterstain the nuclei. p values were determined using the paired Student *t*-test. Scale bars, 10 µm (**H**), 150 µm (**A** and **B**), 250 µm (**F–F"** and **J**). pa, pharyngeal arch; h, heart; d-ot, distal outflow tract; p-ot, proximal outflow tract; rv, right ventricle; lv, left ventricle.

The following figure supplements are available for figure 3:

**Figure supplement 1**. Dual Injection of EdU and BrdU.

**Figure supplement 2**. Time lapse images of 3-D matrigel PA2 culture.

**Figure supplement 3**. Explant culture of PA1 and PA2 dissected from Nkx2.5GFP embryo.

(*Figure 3—figure supplement 2*). The migrated RFP⁺ cells became contractile by day 4 (*Figure 3—figure supplement 2*; *Videos 1 and 2*). Although the range of their migration was somewhat limited, likely due to the extracellular environment of Matrigel, RFP⁺ cells continued to expand in the PA2 and differentiated into cardiomyocytes. When cultured in an uncoated dish, RFP⁺ cells expanded in the PA2 and progressively migrated out of the arch in a unidirectional fashion soon after being attached to the surface. However, no beating activity was observed on day 2 (*Figure 3F*). They migrated out of the arch progressively soon after being attached to the surface (*Figure 3F*). The migrated RFP⁺ cells started to spontaneously contract by day 4 (*Figure 3F'*). The expansion and migration of RFP⁺ cells appear to continue over 2 weeks and the beating area was correspondingly expanded (*Figure 3F''*; *Video 3*). The beating cells were positive for EdU, Nkx2.5, and cTnT/α-Actinin and exhibited a periodic $Ca^{2+}$ oscillation pattern similar to that of adult cardiomyocytes (*Figure 3G–I*). To ascertain that the resulting cardiac cells were not derived from contamination from the adjacent OT, we isolated PAs from *Nkx2.5^GFP* embryos (*Biben et al., 2000*) that express GFP in cardiac cells including the OT, but not in the PAs. Freshly dissected PA2s did not contain GFP⁺ cells, but cells expressed GFP 2–3 days after migration from the arch (*Figure 3—figure supplement 3*). We concluded that RFP⁺ Isl1⁺ cells expand in the PA2 and differentiate into cardiac cells as they migrate out of the arch. It is worth noting that PA1 cells also migrated out from the arch, but they remained GFP⁻and eventually formed myotube-like structures (*Figure 3—figure supplement 3*, *Figure 3J*). This is consistent with the previous finding that pharyngeal mesoderm is also the source of head skeletal muscles (*Nathan et al., 2008*).

## PA2 cells promote the renewal and expansion of CPCs

To determine if the PA2 affects CPC expansion, we isolated Isl1⁺ Nkx2.5⁻ CPCs from ES cells and cultured them with PA2 cells or heart cells derived from E9.0–9.5 embryos or without feeders. The CPCs were obtained by differentiating *Isl1^Cre*; *Ai9* ES cells (*Uosaki et al., 2012*) into Mesp1⁺ precardiac mesoderm (*Kattman et al., 2011*) and purifying RFP⁺ Isl1⁺ Nkx2.5⁻ cells by FACS at day 5 (*Figure 4A*). The RFP⁺ cells spontaneously differentiate and form a sheet of beating cardiomyocytes (*Uosaki et al., 2012*). Strikingly, the co-cultured RFP⁺ cells formed distinct colonies (*Figure 4A*), which were never observed in control culture conditions, and their number was greatly increased over time (*Figure 4B*). The increase was inversely correlated with the appearance of Nkx2.5⁺/cTnT⁺ cells (*Figure 4C*, *Figure 4—figure supplement 1*), indicating PA2 cells promote expansion of Isl1⁺ Nkx2.5⁻ CPCs and suppress their cardiac differentiation. This is unlikely due to the altered 3D environment as the effect was mimicked by PA2 cell-conditioned medium, but not by embryonic heart cells (*Figure 4A,B*, *Figure 4—figure supplement 2*).

To test if the colonies maintain differentiation potential, we established an *Isl1^Cre*; *Ai9*; *Myh6-GFP* ES cell line, in which green fluorescent protein (GFP) is expressed when cells differentiate into cardiomyocytes (*Ieda et al., 2010*). RFP⁺ Isl1⁺

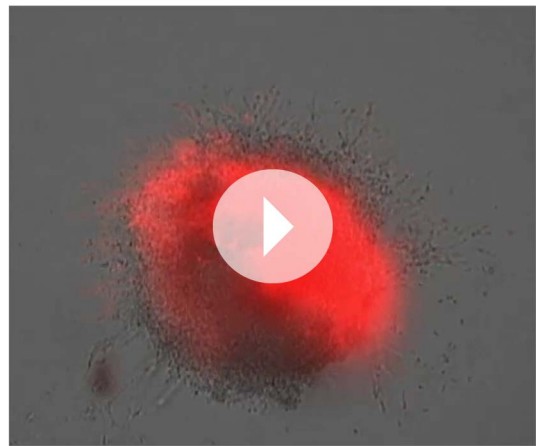

**Video 1**. 3D-cultured PA2 in Matrigel. This video shows expansion and migration of Mesp1 progeny (RFP⁺) in ex-vivo cultured PA2.

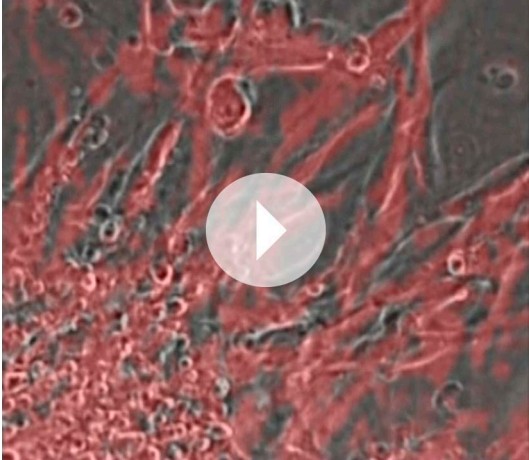

**Video 2**. High magnification of inset in *Video 1*. This video shows RFP⁺ beating cardiac myocytes at a migrating edge of ex-vivo cultured PA2.

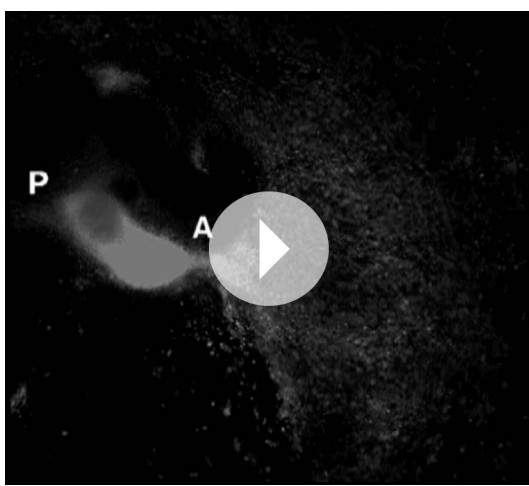

**Video 3**. 2D-cultured PA2. This video shows a beating monolayer of cardiomyocytes derived from RFP+ Mesp1 Progeny in ex-vivo cultured PA2.

Nkx2.5- CPCs were FACS-purified from the line at day 5 of differentiation and co-cultured with PA2 cells to form colonies. In the presence of PA2 cells, the colonies continued to grow without GFP expression. However, when PA2 cells were removed after a week of co-culture, they began to express GFP from day 3 and differentiated into beating cardiomyocytes (*Figure 4D*). These data suggest that PA2 cells provide a cellular environment for the renewal and expansion of Isl1+ Nkx2.5- CPCs and suppress their cardiac differentiation.

## Generation of Mesp1+ cell lineage-specific mosaicism

Although the Cre-mediated conditional deletion revealed a crucial role of Numb and Numbl for the development of Mesp1 progeny in the PA2, it was unclear if the *Numb/Numbl* DKO phenotype reflected the intrinsic role of Numb and Numbl. To address this question, we generated mosaic animals lacking Numb and Numbl in Mesp1 lineages (*Figure 5A*). We first established an *Numb/Numbl* DKO ES cell line, in which conditional deletion of *Numb/Numbl* occurs in Mesp1+ cells with resultant expression of RFP, by directly deriving ES cells from the DKO embryos. The ES cells were injected into host blastocysts of UBI-GFP/BL6 mice (*Schaefer et al., 2001*), which ubiquitously express GFP, to generate chimeras. In this system, *Numb* and *Numbl* deletion occurs in donor-derived Mesp1+ cells, and the DKO cells are traced by RFP expression. Donor-derived RFP+ cells were formed exclusively within Mesp1 lineages and distinguished from GFP+ host cells and RFP-GFP-donor progeny (*Figure 5B–D*). No cells were found double positive for RFP and GFP (*Figure 5C,D*), excluding the possibility of cell fusion, which potentially could rescue the phenotype of DKO cells. Therefore, we used wildtype blastocysts as hosts for further chimera generation. Consistent with the earlier mRNA analysis, Numb was absent in donor-derived RFP+ cells (*Figure 5E*).

The phenotype of DKO chimeras depended on the contribution of the donor RFP+ cells. Major donor contribution caused a phenotype similar to DKO embryos (12.5% 2/16, *Figure 5F,G*). In most cases (87.5% 14/16), the chimeras were indistinguishable from wild-type embryos (*Figure 5H,I*). Donor RFP+ cells were normally populated in the PA2 and contributed to the OT, cardiomyocytes and endocardial cells (*Figure 5J–M*, *Figure 5—figure supplement 1*), indicating deletion of Numb and Numbl did not affect the migration or cardiac differentiation of CPCs.

### *NumbNumbl* DKO CPCs fail to expand in PA2

In control embryos, the Isl1+ Nkx2.5- cells expanded dramatically after E8.5 in the PA2 (*Figure 2C',D,M*). Proliferating cells, marked by PH3, were mostly found within the border of the cluster in control embryos, but were depleted in *Numb/Numbl* DKO embryos (*Figure 6A–C'''*). Likewise, the chimeras formed the Isl1+ cell cluster with PH3+ cells in the PA2 (*Figure 6E*). However, none of the RFP+ Isl1+ Nkx2.5- donor cells was found positive for PH3 in the PA2s of chimeric embryos analyzed at E9.0–9.5 (n = 10, *Figure 6E–I*), implying that Isl1+ Nkx2.5- cells in the PA2 lose their expansion potential in the absence of Numb and Numbl. There was no evidence of apoptosis in *Numb/Numbl* DKO cells in the PA2 (not shown). Consistent with the in vivo data, knockdown of *Numb* and *Numbl* in ES cell-derived Isl1+ Nkx2.5- CPCs, but not in Mesp1+ or Nkx2.5+ CPCs, resulted in a significant reduction of cell proliferation (*Figure 6J*, *Figure 6—figure supplement 1*). Conversely, increased levels of Numb in the Isl1+ Nkx2.5- CPCs promoted their proliferation (*Figure 6K*). These suggest that Numb and Numbl are cell-autonomous factors regulating the expansion of Isl1+ Nkx2.5- cells in the PA2.

## Discussion

Through the use of mouse genetics, lineage-specific mosaicism, embryonic stem cell systems, and ex-vivo organ culture, we have shown the existence of an undifferentiated and expansive population

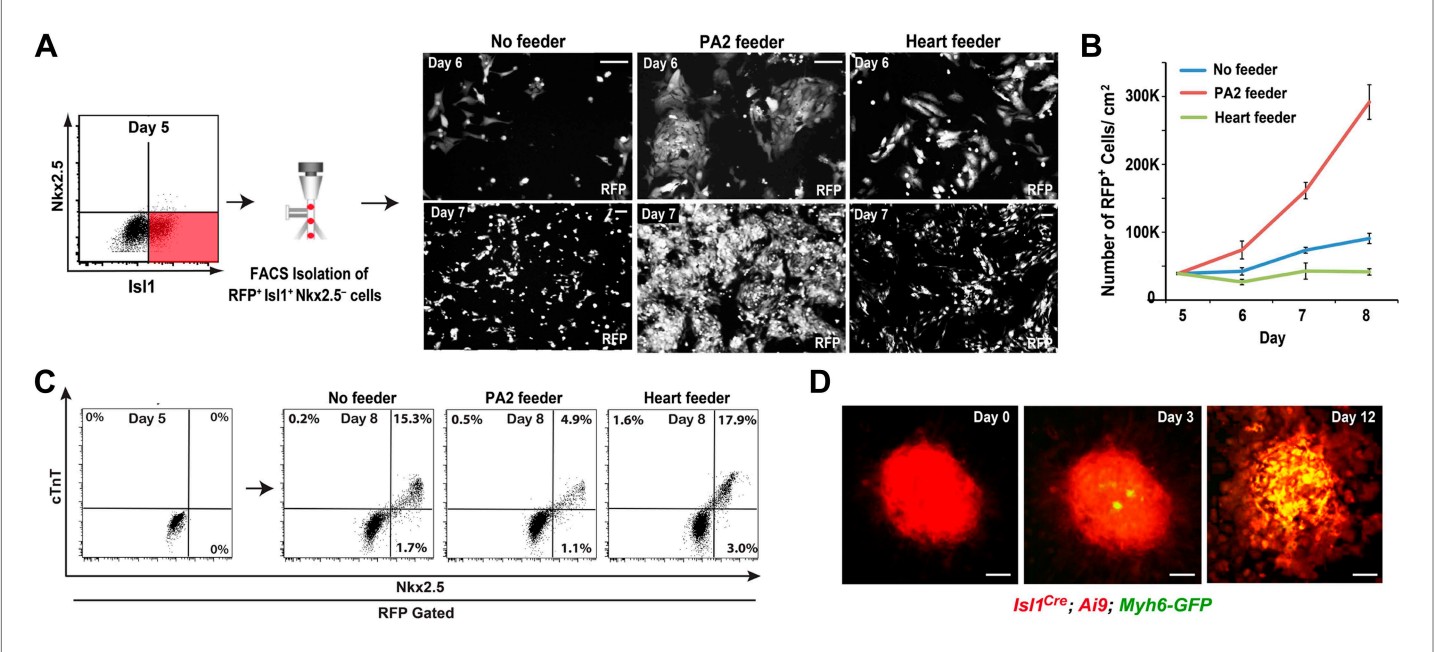

**Figure 4**. PA2 cells promote CPC expansion and suppress cardiac differentiation. (**A**) FACS-purification of RFP+ Isl1+ Nkx2.5− CPCs induced from ES cell-derived precardiac mesoderm and their culture with no, embryonic PA2, or embryonic heart feeders. Images show RFP+ cells at day 6 and day 7. Isl1Cre; Ai9 ES cells were used to purify the CPCs at day 5 of cardiac differentiation, when Nkx2.5 is not expressed in Isl1+ CPCs. (**B**) Quantification of numbers of RFP+ cells cultured with no, embryonic PA2, or embryonic heart feeders. Data are mean ± SD; n = 3. (**C**) FACS plot of RFP+ Isl1+ Nkx2.5− CPCs differentiating into Nkx2.5+/cTnT+ cells with no, embryonic PA2, or embryonic heart feeders, determined at day 8. (**D**) Time-lapse images of Isl1+ Nkx2.5− CPC colony showing cardiac differentiation after removal of PA2 cells, indicated by GFP expression driven by Myh6 promoter. Scale bars, 50 µm.

The following figure supplements are available for figure 4:

**Figure supplement 1**. Percentages of Nkx2.5+ or cTnT + cells cultured with No, PA2, or embryonic heart feeders at day 8.

**Figure supplement 2**. PA2 conditioned medium mimics PA2 co-culture.

of CPCs and their microenvironment during development (**Figure 7**), which may provide a stem cell-niche paradigm in cardiovascular biology. Given that the heart grows rapidly in size and cell number at E8.5–10.5, the CPCs may serve as a renewable source to supply the number of cardiac cells needed to sustain the ensuing heart growth.

Although highly heterogeneous, CPCs are considered as multipotent cells that are destined to become heart cells. However, it remains unclear if they are capable to self-renew without differentiation. In mice, their precursors are identified as early as E5.75 by expression of the T-box transcription factor Eomesodermin in the epiblast (**Russ et al., 2000**) and specified to Mesp1+ cells at the onset of gastrulation at E6.5 (**Costello et al., 2011**). Mesp1+ cells are further specified to CPCs, vascular progenitors and some of the head mesenchyme that contribute to the entire heart, vasculature and subsets of head muscle cells, respectively (**Saga et al., 2000**). CPCs giving rise to the RV migrate through the OT from the SHF and express the transcription factors Isl1, Mef2c or Nkx2.5. These factors, however, are also expressed in neighboring cells including pharyngeal ectoderm and endoderm, foregut endoderm, neural progenitors and neural crest cells that are not originated from mesoderm (**Lints et al., 1993**; **Edmondson et al., 1994**; **Cai et al., 2003**), making it difficult to precisely discern CPCs. Moreover, it is unclear when and where CPCs are specified from Mesp1+ cells or their progeny. Thus, we traced Mesp1 lineages and examined the expression of CPC markers in Mesp1 progeny. Unexpectedly, heart cells rarely proliferated during early cardiogenesis, implying the existence of an alternate cell source for the ongoing growth of the heart. Indeed, we found a proliferative cluster of Mesp1+ cell-derived Isl1+ Nkx2.5−cells in the PA2—directly linked to the Isl1+ Nkx2.5+ OT of the growing heart—that migrated to the heart and became heart cells. The Isl1+ Nkx2.5− CPCs continued

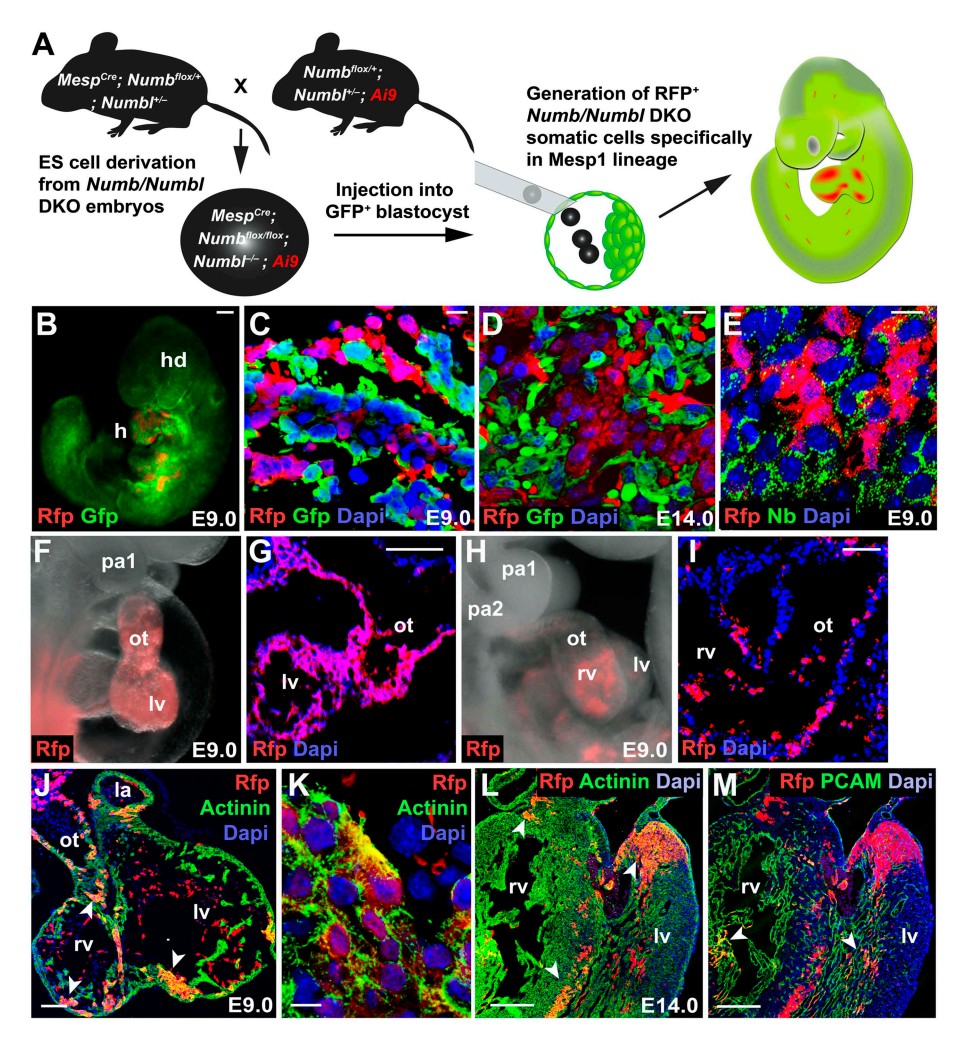

**Figure 5**. Generation of Mesp1 lineage-specific somatic cells lacking Numb and Numbl in vivo. (**A**) Scheme for generation of RFP+ *Numb/Numbl* DKO cells in Mesp1 lineage. In this study, three independent sets of blastocyst injection were carried out and 16 chimeras were obtained from 132 embryos. (**B**–**E**) Chimeric embryos at E9.0 (**B**) generated with GFP+ host cells. *Numb/Numbl* DKO cells are shown in red. Sections were made transversely through cardiac region (**C**–**E**) and immunostained with RFP and GFP (**C** and **D**) or RFP and Numb (**E**) antibodies at corresponding stages. (**F**–**I**) Lateral views of chimera and sections, focused on PA and heart, showing contribution of RFP+ cells. Major contribution of RFP+ cells causes a phenotype similar to *Numb/Numbl* DKO embryos (**F** and **G**). (**J**–**M**) Confocal images of heart transverse sections at indicated stages. α-Actinin and PCAM are properly expressed in RFP+ cells (arrowheads). RFP+ area in (**J**) is enlarged in (**K**). Dapi (blue) was used to counterstain the nuclei. Scale bars, 10 μm (**C**, **D**, **E**, **K**), 100 μm (**B**, **G**, **I**, **J**), and 200 μm (**L** and **M**). a, anterior; p, posterior; hd, head; h, heart; pa, pharyngeal arch; ot, outflow tract; rv, right ventricle; lv, left ventricle; la, left atrium.

The following figure supplements are available for figure 5:

**Figure supplement 1**. Numb/Numbl DKO cells are normally specified into OT cells.

to expand in vivo, ex vivo, and in vitro without cardiac differentiation in the PA2, suggesting that the CPCs may serve as a renewable cell source for the developing heart. This cell renewal system may provide a parsimonious and efficient way to quickly generate a large number of cells used to build the heart during embryogenesis, which can be advantageous over local proliferation of differentiated cardiac cells. Further characterization of the CPCs will be necessary to provide quantitative information on their cellular contribution to the developing heart. Recent studies suggested that subsets of

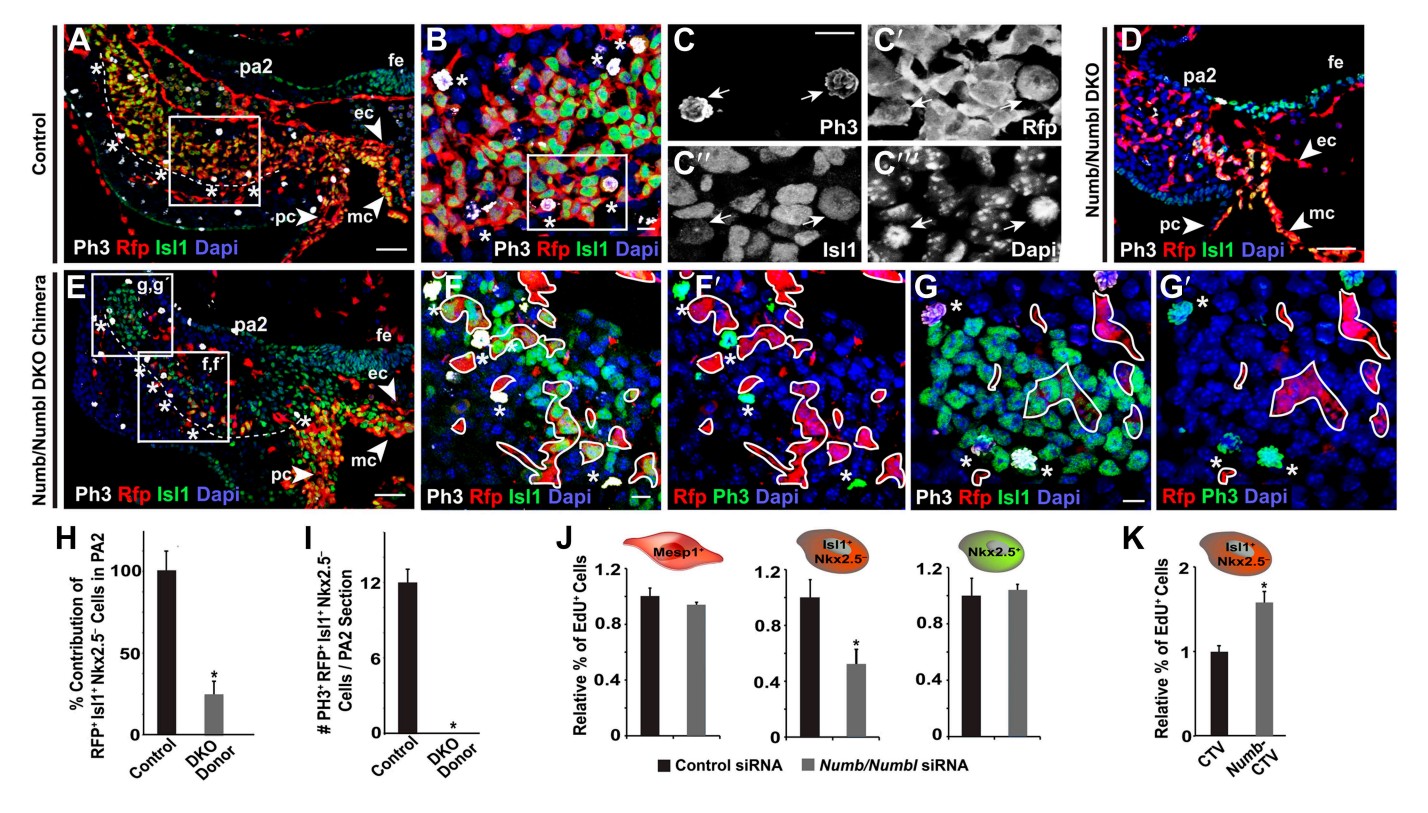

**Figure 6**. Numb and Numbl are required for proliferation of Mesp1+ progenitor-derived Isl1+ Nkx2.5– cells in PA2. (**A–G'**) Confocal images of PA2 sections of control (**A–C'''**), *Numb/Numbl* DKO (**D**), and chimeric (**E–G'**) embryos, immunostained with PH3, RFP, Isl1 antibodies. Asterisks indicate PH3+ RFP+ Isl1+ (triple positive) cells located in outer layers of RFP+ Isl1+ cell population (outlined, **A** and **E**). Boxed areas in (**A**), (**B**), and (**E**) are shown in higher magnification in (**B**), (**C–C'''**), and (**F–G'**), respectively. No PH3+ cells are found in RFP+ cells (asterisks, **F–G'**). RFP+ cells were outlined in white (**F–G'**). (**H** and **I**) Percentage of donor-derived Isl1+ Nkx2.5– cells in PA2 is shown in (**H**) and number of PH3+ RFP+ Isl1+ cells per 12-micron PA2 section is shown (**I**). Data are mean ± SD; n = 10; *p<0.05. (**J** and **K**) Relative percentage of EdU+ cells in ES cell-derived Mesp1+ progenitor, Isl1+ Nkx2.5– CPCs or Nkx2.5+ CPCs transfected with control or *Numb/Numbl* DKO siRNA (**J**) or Isl1+ Nkx2.5– CPCs transfected with control (CTV) or Numb overexpression construct (CTV-Numb) (**K**). Data are mean ± SD; n = 3; *p<0.05. The Mesp1+, Isl1+ Nkx2.5–, or Nkx2.5+ cells were FACS-purified from day 4 *Mesp1^Cre; Ai9*, day 5 *Isl1^Cre; Ai9*, or day 6 *Nkx2.5^GFP* ES cells, respectively. Dapi (blue) was used to counterstain the nuclei. p values were determined using the paired Student *t* test. Scale bars, 10 μm (**B**, **C**, **F**, **G**). 50 μm (**A**, **D**, **E**). fe, foregut endoderm; pa, pharyngeal arch ec, endocardial layer; mc, myocardial layer; pc, pericardial layer.

The following figure supplements are available for figure 6:

**Figure supplement 1**. Knockdown efficiency of Numb siRNA and Numbl siRNA.

head and cardiac muscles share their progenitors (*Lescroart et al., 2010*). While the progenitor has not been identified, it will be interesting to investigate if this progenitor is derived from Isl1+ Nkx2.5– cells.

PAs are transient, segmented bulges that appear on the craniolateral side of developing embryos (*Grevellec and Tucker, 2010*). Each PA is composed of a mesodermal core and neural crest cell–derived mesenchymal cells that are surrounded by ectoderm outside and endoderm inside. At E8.5, the PA1—positioned most cranially among PAs—is structurally distinct, while the PA2 becomes noticeable after E9.0. Mesp1-derived Isl1+ Nkx2.5– cells proliferate in PA2s and initiate expression of Nkx2.5+ soon after exiting the arch. This suggests that PA2s function as a microenvironment to maintain the CPCs in an undifferentiated and expanding state and likely contain cells secreting paracrine factors that control the CPC number and fate. In fact, numerous signaling molecules are secreted from PA endoderm, ectoderm, and mesenchyme including Wnts, bone morphogenetic proteins, sonic hedgehog, and fibroblast growth factors (*Rochais et al., 2009*), and dysregulation of these signals is often associated with OT/RV defects (*Frank et al., 2002*; *Stottmann et al., 2004*; *Washington Smoak*

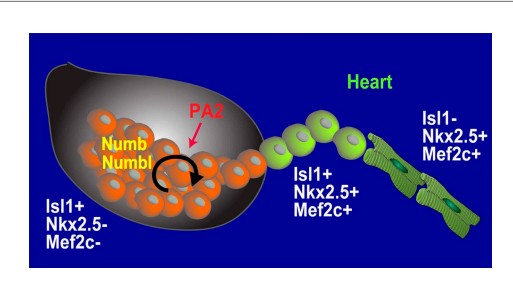

**Figure 7**. Model for Renewal and Niche of Mesp1+ progenitor-derived CPCs.

et al., 2005; Kwon et al., 2007). Thus, it will be important to identify the extrinsic factors and cell types that provide key signals for the CPC maintenance and PA2 development.

Although the heart phenotype (hypoplastic OT/RV) caused by *Numb/Numbl* DKO may not result entirely from CPC depletion in the PA2, our findings together with published literatures suggest that the CPCs in the PA2 might be a major source of cells contributing to the OT/RV. For instance, the SHF—giving rise to the entire OT/RV—is located in PAs (*Kelly et al., 2001*; *Rochais et al., 2009*) and we found that proliferating cells are present predominantly in the PA1 and PA2 and rarely detected in the rest of the PAs. Our ex-vivo culture further showed that PA2 cells robustly differentiate into cardiac cells, whereas PA1 cells appear to give rise to myotubes without cells expressing Nkx2.5. It is also worth noting that nearly all, if not all, embryos showing severely hypoplastic OT/RV exhibit hypoplastic PA2s (*Srivastava et al., 1997*; *Gottlieb et al., 2002*; *Cai et al., 2003*; *Kwon et al., 2007*, *2009*).

Numb and Numbl are highly conserved proteins that participate in cell fate determination by mediating asymmetric division, endocytosis and recycling of specific proteins, ubiquitination and cell migration (*Santolini et al., 2000*; *Cayouette and Raff, 2002*). Classic studies of *Drosophila* demonstrated Numb's spatio-temporal segregation to one pole of the mitotic cell as the primary mechanism by which cell fate is determined in single organ precursors (*Uemura et al., 1989*). In mammals, Numb and Numbl are required for the self-renewal of neural progenitors to maintain their number during development; while in the other settings they promote a neuronal fate by neural progenitor specification (*Petersen et al., 2002*, *2004*). Similarly, Numb and Numbl were required for the renewal of CPCs during cardiogenesis, suggesting a conserved role in progenitor maintenance. It is unlikely that disruption of the yolk sac or angiogenesis contributed to the restriction of cardiac growth because there was no discernable difference in the phenotype of embryos or yolk sacs at E8.5. Furthermore, the results from somatic mosaicism demonstrated that *Numb/Numbl* DKO CPCs were unable to proliferate in normal PA2 environment, suggesting a cell-autonomous role of Numb and Numbl. Curiously, the deletion of Numb and Numbl in CPCs appears to affect the growth of neighboring PA2 cells as well. This suggests that Numb and Numbl may also influence CPC renewal and expansion by regulating PA2 development in a non-cell autonomous manner.

Numb and Numbl may affect CPC renewal and expansion via Notch, a conserved transmembrane receptor, given that Numb and Notch are mutually antagonistic (*Schweisguth, 2004*). In fact, Notch1 deficiency causes CPC expansion in the OT (*Kwon et al., 2009*). The expansion of CPCs is at least in part mediated by accumulation of the Wnt signaling effector β-catenin that are negatively regulated by membrane Notch (*Kwon et al., 2011*). Membrane Notch appears to require Numb and Numbl for the negative regulation of β-catenin (*Kwon et al., 2011*; *Andersen et al., 2012*), suggesting Numb and Numbl may be essential regulators of Notch and Wnt signaling during CPC development. With our current study, it will be necessary to re-examine the roles of Notch and Wnt signals at the level of renewing CPCs in the PA2.

## Materials and methods

### Mouse genetics and ES cell culture

*Numb/Numbl* DKO mouse embryos were generated by mating *Mesp1^Cre^; Numb^flox/+^; Numbl^−/+^* with *Numb^flox/flox^; Numbl^−/+^; Ai9* mice. Embryos were harvested from E7.0–10.0 and genotyped as described (*Saga et al., 1999*; *Petersen et al., 2002*; *Madisen et al., 2010*). Nkx2.5^GFP^ mice were used for ex-vivo culture (*Biben et al., 2000*). For ES cell work, ES^Mesp1−Cre; Ai9^ cells (this work), ES^Mesp1−Cre; Numb flox/flox or flox/+; Numbl−/+ or −/−; Ai9^ cells (this work), ES^Isl1−Cre; Ai9^ cells (*Uosaki et al., 2012*), ES^Nkx2.5−GFP^ cells (*Hsiao et al., 2008*), and ES^Isl1−Cre; Ai9; Myh6−GFP^ cells (this work) were used. ES^Mesp1−Cre; Ai9^, ES^Mesp1−Cre; Numb flox/flox or flox/+; Numbl−/+ or −/−; Ai9^ or ES^Isl1−Cre; Ai9; Myh6−GFP^ were derived from mice harboring *Mesp1^Cre^; Ai9, Mesp1^Cre^; Numb^flox/flox or flox/+^; Numbl^+/− or −/−^; Ai9* or *Isl1^Cre^; Ai9; Myh6-GFP*, respectively. ES cells were maintained on gelatin-coated

dishes in maintenance medium (Glasgow minimum essential medium with 10% fetal bovine serum and 1000 U/ml ESGRO (Millipore, Billerica, MA), Glutamax (Life Technologies/Thermo Fisher Scientific K.K., Waltham, MA), sodium pyruvate, MEM non-essential amino acids) and CPCs were induced with activin A, BMP4, and VEGF (R&D Systems, Minneapolis, MN) (*Kattman et al., 2011*; *Cheng et al., 2013*).

## Ex-vivo culture and PA2 co-culture

For explant culture, PA1s or PA2s were carefully dissected from embryos at E9.0–9.5. Absence of contamination from the adjacent outflow tract was confirmed by absence of Nkx2.5[+] cells. The explants were cultured in standard serum free medium supplemented with ascorbic acid at 37°C in 5% $CO_2$. For 3D cultures, the PA explants were cultured in Matrigel. For PA2 cell co-culture, PA2s were isolated from E9.0–9.5 *Isl1^{Cre}*; *Ai9* embryos and cultured in gelatin-coated plate with PA media (DMEM: F-12, 7.5% FBS, 1X penicillin-streptomycin, 1X Glutamax). RFP[−] cells from PA's with significant outgrowth were isolated and further passaged. The PA2-derived or control (embryonic heart) cells were plated at a density of 10,000–50,000 cells/cm$^2$ in multi-well plates and ES cell-derived CPCs were plated on top of the PA2 cells at a density of ~10,000 cells/cm$^2$ in SFD medium.

## Generation of Mesp1 lineage-specific somatic cells

For *Numb/Numbl* DKO ES cell derivation, 3- to 4-week-old *Numb^{flox/flox}*; *Numbl^{−/+}*; *Ai9* female mice were super-ovulated and mated with *Mesp1^{Cre;} Numb^{flox/+}*; *Numbl^{−/+}* males. Blastocysts were flushed at E3.5 and cultured to establish mouse ES cell lines (*Ying et al., 2008*). *Numb/Numbl* DKO ES cell lines were identified by genotyping and karyotyped to select suitable lines for the production of chimera. *Numb/Numbl* DKO ES cells were injected into the UBI-GFP/BL6 or wildtype blastocysts to generate chimera, which were transferred to E0.5–1.5 pseudo-pregnant recipient mothers. Chimeric embryos were harvested and analyzed at E7.5, 9.0, 14.0.

## siRNA, constructs, and transfection

For *Numb* and *Numbl* knockdown experiments, Numb/Numbl ON-TARGETplus SMARTpool siRNA (L-046935/L-046983) or scrambled siRNA (Dharmacon/Thermo Fisher Scientific K.K.) was used at 100 nM for cell transfection (*Kwon et al., 2011*). For Numb overexpression, full-length *Numb* cDNA was cloned into CTV vector (*Xiao et al., 2007*) and used to increase Numb levels by transfecting in Cre-expressing cells. Cells were transfected with Lipofectamine LTX or Lipofectamine 2000 (Life Technologies) in single-cell suspensions.

## EdU labeling, immunohistochemistry, and microscopy

For EdU pulse-tracing experiments, 10 mM EdU was injected at 0.075 mg per gram bodyweight intra-peritoneally to pregnant mice at E9.0. Embryos were harvested at 2, 4, 8, and 32 hr post EdU injection. We used the Click-it EdU kit (Life Technologies) for EdU detection followed by immunostaining with primary and secondary antibodies. For the dual injection experiment, EdU and BrdU (300 µl, 10 mM each) were sequentially injected at a 4-hr interval. Embryos were harvested 4 hr post BrdU injection (8 hr of EdU), fixed, and sectioned (12 µm thickness). Antigen retrieval was performed with microwave for 20 min in 10 mM EDTA solution. The section was epimerized with 0.2% PBS Triton X for 15 min and stained with the Click-it EdU kit. The resulting sections were washed and incubated with anti BrdU antibody in incubation buffer (Roche BrdU Labeling and Detection Kit I). Anti-Mouse Ig-Fluorescein was used as secondary antibody. For microscopy, embryos were fixed in 4% paraformaldehyde overnight and then 30% sucrose, and then embedded in OCT, sectioned and stained using standard protocols. Antibodies used were: goat α-Islet1 (1:200; R&D), goat α-PECAM (1:200; R&D), mouse α-Islet1 (1:200; Developmental Studies Hybridoma Bank, Iowa City, IA), rabbit α-MEF2c (1:200; Cell Signaling Technology, Danvers, MA), rabbit α-RFP (1:400; Clontech Laboratories, Inc., Mountain View, CA), rabbit α-Numb (pre-absorbed, 1: 500; Abcam, Cambridge, MA or from Dr. Zhong), goat or rabbit anti-β1 integrin (1:400; R&D or 1:1000; Abcam), mouse α-sarcomeric Actinin (1:500; Sigma-Aldrich, St. Louis, MO), goat α-Nkx2.5 (1:20; Santa Cruz Biotechnology, Dallas, Texas), mouse α-PH3 (1:500; Abcam), rabbit α-GFP (1:400; Abcam), goat α-GFP (1:200; R&D), rabbit α-caspase 3 (Abcam). Alexa Fluor secondary antibodies (Life Technologies) were used for secondary detection and confocal images acquired with a Zeiss LSM 510 Meta confocal microscope using Zen acquisition software.

## Flow cytometry and time-lapse imaging

For flow cytometry, cells were dissociated and analyzed with Accuri C6 Flowcytometer (BD Biosciences, San Jose, CA) and FlowJo software (TreeStar, Ashland, OR). For intracellular-flow cytometry, cells were

stained with indicated antibodies after dissociation as previously described (*Uosaki et al., 2011*). For FACS, dissociated cells were resuspended in PBS containing 0.1% FBS, 20 mM Hepes and 1 mM EDTA and sorted with FACSAria II (BD Biosciences) and SH800 sorter (Sony Biotechnology, Japan). Time-lapse imaging was done with a BZ9000 All-in-One Fluorescence microscope (Keyence, Japan).

## Ca2$^+$ transient measurement and analysis

PA2 explants were incubated with 3 µM Fura-2 AM (Invitrogen, Molecular Probes, Carlsbad CA) for 20 min at 37°C. After washing and de-esterification for 20 min the explants were placed in an imaging chamber and electrical field stimulated at 1 Hz, 37°C, pH 7.4 with Ca2$^+$. The change in intracellular Ca2$^+$ was measured with an inverted fluorescence microscope (TE2000, Nikon, Japan) and Myocam (IonOptix, Milton, MA) by Fura-2 AM fluorescence intensity ratio at 340 nm and 380 nm.

## Statistical analyses

Differences between groups were examined for statistical significance using the paired Student's *t* test. A p-value <0.05 was regarded as significant. Error bars indicate standard error of the mean.

## Acknowledgements

We thank YN Jan (UCSF) for generously providing *Numb^flox^* and *Numbl* knockout mice. The authors thank Kwon and Kass laboratory members for helpful discussions, the Johns Hopkins University Transgenic Core (A Lawler and C Hawkins) for technical assistance, and L Sprague for help with time-lapse experiments. CK was supported by the Magic That Matters Fund, grants from NHLBI/NIH (R01HL111198, R00HL092234), and MSCRF. PA was supported by the Lundbeck foundation (Denmark). HU was supported by Japan Society for the Promotion of Science. DAK was supported by Fondation Leducq, Muscular Dystrophy Association, and HL:107153, and PPR was supported by the Max Kade Foundation.

## Additional information

### Funding

| Funder | Grant reference number | Author |
| --- | --- | --- |
| Magic That Matters Fund | | Chulan Kwon |
| National Heart, Lung, and Blood Institute | R01HL111198 | Chulan Kwon |
| National Heart, Lung, and Blood Institute | R00HL092234 | Chulan Kwon |
| Maryland Stem Cell Research Fund | 2012-MSCRFE-0127-00 | Chulan Kwon |
| Lundbeck foundation | | Peter Andersen |
| Fondation Leducq | | David A Kass |
| Max Kade Foundation | | Peter P Rainer |
| Muscular Dystrophy Association | | David A Kass |
| National Institutes of Health | HL:107153 | David A Kass |

The funders had no role in study design, data collection and interpretation, or the decision to submit the work for publication.

### Author contributions

LTS, PA, Conception and design, Acquisition of data, Analysis and interpretation of data, Drafting or revising the article; HU, PPR, G-sC, Conception and design, Acquisition of data, Analysis and interpretation of data; LF, D-iL, Acquisition of data, Analysis and interpretation of data; WZ, RPH, Analysis and interpretation of data, Contributed unpublished essential data or reagents; DAK, Conception and design, Analysis and interpretation of data; CK, Conception and design, Analysis and interpretation of data, Drafting or revising the article

### Ethics

Animal experimentation: All the animals were treated humanely according to AALAC and NIH guidelines (Johns Hopkins Medical Institutions are fully accredited by the AALAC) and the current study

was conducted under the animal protocol (MO10M444) approved by the Animal Care and Use Committee at Johns Hopkins University.

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
