## [Decision Letter]

Thank you for sending your work entitled “Precardiac Deletion of Numb and Numblike Reveals Renewal of Cardiac Progenitors” for consideration at *eLife*. Your article has been favorably evaluated by a Senior editor and 3 reviewers, one of whom is a member of our Board of Reviewing Editors.

The Reviewing editor and the other reviewers discussed their comments before reaching this decision, and the Reviewing editor has assembled the following combined comments to help you prepare a revised submission.

This manuscript represents a highly skilled use of mouse genetics, stem cell approaches and organ culture to dissect some of the early events that occur during cardiac development. The authors deploy a dazzling set of genetic manipulations to investigate the role of Numb and Numb-like in cardiac development. The authors conclude that Nb/Nbl function largely cell-autonomously in PA2 cells to modulate cardiac precursor proliferation and repress differentiation. The chimera experiments clearly indicate that there is a cell-autonomous role for Numb and Numb-like, at the very least in regulating proliferative capacity of Isl1^+^ progenitors. Overall this is an important area of study and the work is very rigorously done, with deep penetration into the processes examined.

The authors address a long-standing question of how cardiac progenitors get added to the developing heart, which expands rapidly but without very much cell division once differentiated. They find that a pool of cardiac progenitors derived from Mesp1-expressing cells in the second pharyngeal arch are maintained in a proliferative and expansive state without further differentiation and give rise to progressively more differentiated cells once they migrate out of the arch. It is this expanding population that provides the cells for the developing heart and this is shown elegantly by organ culture and complimented by flow studies in ES-derived progenitors with markers representing each stage of differentiation. By generating mice double null for the redundant Numb and Numb-like proteins, they show that these progenitors are not maintained in the absence of Numb. Cre-based lineage tracing experiments in these mice revealed that the problem was in expansion, and elegant chimera studies demonstrated that the defect was a cell-autonomous one involving Numb in the cardiac progenitors. They took the work a step further and show that β-integrin expression in the cardiac progenitors was dependent on Numb and that deletion of β-integrin in the cardiac progenitors resulted in their lack of expansion.

Overall, this is an exceptional body of work that is extremely thorough, the data are of very high quality, and the conclusions are largely supported by the data shown. While the authors conclusively demonstrate that the microenvironment in the second pharyngeal arch maintains the progenitors in an undifferentiated and expansive state, it remains unknown what that signal is. It is commendable that the authors demonstrate that β-integrin is involved in receiving that signal, but future work will be necessary to reveal if one or a combination of Wnt/Notch/Fgf signals are the critical ones. I believe it is acceptable that that signal is not determined in this already large body of work.

There are however, some specific points the authors need to address to bring this work up to a level required for publication in *eLife*.

1) A major concern is that while the phenotype in PA2 cells is evident, the authors have not considered other regions of expression of Nb/Nbl. In a publication in Development, another group has shown that they are also important in other precardiac and cardiac populations. How can the effects in the PA2 be directly linked or predominantly associated with the resulting phenotype? This should be at least discussed and the authors could provide evidence for a minor role elsewhere in the embryo if it is available.

2) The EdU experiments are somewhat difficult to interpret, as there are no controls to show the perdurance of the pulse. If available, an earlier pulse showing no labeling after a certain time should be shown to demonstrate perdurance.

3) The explant experiments are suggestive of the potential for PA2 cells to differentiate into cardiac lineages, but it would be important to have a live marker such as Nkx2-5::EGFP to ascertain that no slight contamination from the adjacent outflow tract was not included, or at least immunostaining on representative explanted PA2 segments. Is this available or can the authors comment on how they excluded this possibility.

4) The significance of the co-culture experiments in Figure 4 is unclear. The PA2 cells promote the formation of aggregates, and appear to reduce differentiation potential. Is this a direct effect of the PA2 cells on differentiation, or is the altered 3D environment responsible?

5) The link to B-integrin is not readily apparent and appears to be solely based on selecting this protein for observation and demonstrating that its expression is absent in the DKOs. This aspect of the work seems to be an add-on of available data and does not flow logically from the work described before. The phenotype of deletion of β-integrin is similar to that of the DKOs, but it is not possible to determine if β-integrin is a key mediator of the function of Numb/NBL. In the absence of a rescue, this set of results is purely speculative and should be either removed, or the conclusions toned down.

6) For the general readership the flow of the text is difficult to read. This may be due to the extensive list of markers, techniques and approaches, which make it hard to extract the key findings. Perhaps the authors could deal with this issue by expanding a bit on what they have found and why it is important at the end of each section. They do this to some extent but it is very terse. There is room to summarize the finding being tested and the implications of the result.

---

## [Author Response]

*1) A major concern is that while the phenotype in PA2 cells is evident, the authors have not considered other regions of expression of Nb/Nbl. In a publication in Development, another group has shown that they are also important in other precardiac and cardiac populations. How can the effects in the PA2 be directly linked or predominantly associated with the resulting phenotype? This should be at least discussed and the authors could provide evidence for a minor role elsewhere in the embryo if it is available*.

Although the main conclusion of this study is the existence of renewing CPCs and their microenvironment (PA2) during heart development, we agree that it is important to discuss why this population is distinct and how important it is. While the phenotype (hypoplastic OT/RV) caused by *Numb/Numbl* DKO may not result entirely from CPC depletion in the PA2, our findings together with published literatures suggest that the CPCs in the PA2 might be a major source of cells contributing to the OT/RV. For instance, the SHF – giving rise to the entire OT/RV – is located in PAs (19; 31) and we found that proliferating cells are present predominantly in the PA1 and PA2, and rarely detected in the rest of the PAs (Figure 3). Our ex-vivo culture further showed that PA2 cells robustly differentiate into cardiac cells, whereas PA1 cells give rise to myotube-like cells without expressing Nkx2.5 (Figure 3—figure supplement 3, Figure 3). It is also worth noting that nearly all, if not all, embryos showing severely hypoplastic OT/RV exhibit hypoplastic PA2s (41; 12; 4; 20; 22). We discussed these points and included new data in the revised manuscript.

We believe that the finding published in Development (Zhao et al. 2014) does not affect our findings/conclusions. While we showed that *Numb/Numbl* deletion in the earliest cardiac progenitors (Mesp1^+^) results in hypoplastic OT/RV with embryonic lethality at E9.5 and affects the renewal of undifferentiated, Nkx2.5 NEGATIVE CPCs, the published study reported morphological defects from E10.5–16.5 after deleting *Numb/Numbl* in Nkx2.5 POSITIVE cardiac cells (induced later than Mesp1^+^ cells). We confirmed with Dr. Wu (senior author of the paper) that the early defects (hypoplastic OT/RV) we described are not present when *Numb* and *Numbl* are deleted later in Nkx2.5^+^ cells. These studies clearly demonstrate that Numb and Numbl function in a spatio-temporally distinct manner during development.

*2) The EdU experiments are somewhat difficult to interpret, as there are no controls to show the perdurance of the pulse. If available, an earlier pulse showing no labeling after a certain time should be shown to demonstrate perdurance*.

To address this concern, we sequentially injected EdU (at E9.5) followed by BrdU injection after 4 hours. The resulting embryos were harvested after 4 hours (8 hours after EdU injection) and analyzed by immunostaining. As shown in Figure 3—figure supplement 1, EdU^+^ cells were mostly negative for BrdU, suggesting the purdurance of EdU is shorter than 4 hours. It’s also worth noting that some EdU^+^ BrdU^+^ cells were present in the PA2, implying their continued proliferation. We believe this finding strengthens our in vivo and in vitro EdU pulse-tracing data in Figure 3.

*3) The explant experiments are suggestive of the potential for PA2 cells to differentiate into cardiac lineages, but it would be important to have a live marker such as Nkx2-5::EGFP to ascertain that no slight contamination from the adjacent outflow tract was not included, or at least immunostaining on representative explanted PA2 segments. Is this available or can the authors comment on how they excluded this possibility*.

We utilized Nkx2.5-GFP mice as suggested and showed that freshly dissected PA2 explants do not contain Nkx2.5^+^ cells (Figure 3—figure supplement 3).

*4) The significance of the co-culture experiments in*
Figure 4
*is unclear. The PA2 cells promote the formation of aggregates, and appear to reduce differentiation potential. Is*
*this a direct effect of the PA2 cells on differentiation, or is the altered 3D environment responsible?*

The PA2 cell effect (colony formation with reduced differentiation) was not observed when the cardiac progenitor cells were co-cultured with the same density of embryonic heart cells (Figure 4, Figure 4—figure supplement 1), suggesting the effect is unlikely due to the altered 3D environment. We further found that culturing the progenitor cells in PA2 cell-conditioned medium significantly increased their number and decreased their cardiac differentiation (Figure 4—figure supplement 2). These data suggest that PA2 cells have a direct role for the effect.

*5) The link to B-integrin is not readily apparent and appears to be solely based on selecting this protein for observation and demonstrating that its expression is absent in the DKOs. This aspect of the work seems to be an add-on of available data and does not flow logically from the work described before. The phenotype of deletion of β-integrin is similar to that of the DKOs, but it is not possible to determine if β-integrin is a key mediator of the function of Numb/NBL. In the absence of a rescue, this set of results is purely speculative and should be either removed, or the conclusions toned down*.

We agree with the comment and removed the β-integrin data as suggested.

*6) For the general readership the flow of the text is difficult to read. This may be due to the extensive list of markers, techniques and approaches, which make it hard to extract the key findings. Perhaps the authors could deal with this issue by expanding a bit on what they have found and why it is important at the end of each section. They do this to some extent but it is very terse. There is room to summarize the finding being tested and the implications of the result*.

We emphasized/expanded key findings and added a sub-conclusion at each step/section as suggested.